# Complementary Role of Ultrasound and Clinical Features in Assessing Carpal Tunnel Syndrome Severity: A Cross-Sectional Study

**DOI:** 10.3390/diagnostics15232985

**Published:** 2025-11-24

**Authors:** Daniela Nicoleta Popescu, Claudiu Costinel Popescu, Oana Morari, Natalia Blidaru, Alina Dima, Ioana Adriana Catanoiu, Alice Rakoczy, Ioana Otobic, Magda Ileana Parvu, Catalin Codreanu, Luminita Enache

**Affiliations:** 1Doctoral School of Medicine, “Carol Davila” University of Medicine and Pharmacy, 020021 Bucharest, Romania; daniela-nicoleta.popescu@drd.umfcd.ro; 2Department of Rheumatology, Colentina Clinical Hospital, 020125 Bucharest, Romania; 3Department of Rheumatology, “Carol Davila” University of Medicine and Pharmacy, 050474 Bucharest, Romania; 4“Dr. Ion Stoia” Clinical Center of Rheumatic Diseases, 020983 Bucharest, Romania; 5Department of Neurology, Colentina Clinical Hospital, 020125 Bucharest, Romania; 6Department of Clinical Neurosciences, “Carol Davila” University of Medicine and Pharmacy, 050474 Bucharest, Romania

**Keywords:** carpal tunnel syndrome, ultrasound, nerve conduction study

## Abstract

**Background/Objectives**: The goal of this study was to assess the correlation between ultrasound measurements and nerve conduction study (NCS)-defined carpal tunnel syndrome (CTS) severity and to explore clinical and demographic factors associated with CTS severity in a sample of Romanian patients. **Methods**: We prospectively evaluated consecutive patients with clinically diagnosed CTS. All patients underwent standardized clinical assessment, ultrasonographic examination of the median nerve, and NCS. CTS severity was graded electrophysiologically (three-level scale), and associations with demographic, clinical, and ultrasound parameters were examined using univariate analyses and multivariable generalized estimating equation (GEE) models to account for within-patient clustering. **Results**: Among 193 CTS hands (100 patients, mean age 58 years, 93% female), electrophysiological severity correlated significantly with several ultrasound and clinical parameters. In multivariable GEE models, the presence of nocturnal symptoms, sensory loss, thenar weakness/atrophy, male sex, larger maximal median nerve cross-sectional area (mCSA), and impaired median nerve mobility were independent predictors of higher NCS-defined severity. Pseudo-R^2^ increased from 0.04 in the core clinical model to 0.25 when ultrasound parameters were included, indicating improved model performance. **Conclusions**: Ultrasound parameters, particularly mCSA and median nerve mobility, together with clinical features, such as nocturnal symptoms, sensory loss, and thenar weakness, are independently associated with NCS-defined CTS severity. These findings support the complementary role of ultrasound alongside NCS in severity grading and highlight its potential to guide timely diagnosis and management.

## 1. Introduction

Carpal tunnel syndrome (CTS) remains the most prevalent form of peripheral nerve entrapment [1,2] and is associated with paresthesia, sensorimotor deficits, and thenar muscle atrophy [3,4,5], which, if left undiagnosed, may progress to significant functional impairment [6,7]. Given its high prevalence and impact on quality of life, there is a pressing need for diagnostic tools that are accessible, easy to use, and cost-effective [8,9].

CTS is traditionally diagnosed through clinical evaluation corroborated by electrodiagnostic testing, specifically nerve conduction studies (NCS) and electromyography (EMG). Over the past decade, neuromuscular ultrasound has emerged as a promising adjunct or alternative diagnostic tool, due to its non-invasive nature, lower cost, and ability to directly visualize median nerve morphology and wrist anatomy. The 2012 guidelines from the American Association of Neuromuscular and Electrodiagnostic Medicine support the use of ultrasound as a complementary tool to electrodiagnosis [10]. Ultrasound is particularly useful in identifying structural causes of median nerve compression, such as flexor tenosynovitis, wrist synovitis, or crystal deposits, and in detecting other potential entrapment sites. Moreover, in a subset of patients with clinical signs of CTS but normal NCS findings, ultrasound may reveal median nerve enlargement, underscoring its unique diagnostic value in such cases [11,12]. Ideally, patients with suspected CTS should undergo all three diagnostic modalities: clinical, ultrasonographic, and electrophysiological, though this is often feasible only in specialized centers. NCS remains the diagnostic gold standard and is routinely recommended before surgery, as it objectively confirms median nerve dysfunction, excludes alternative neuropathies or radiculopathies, and provides prognostically relevant baseline measurements [13,14,15]. However, the limited accessibility and higher cost of NCS in many healthcare settings further underscore the importance of relying on more widely available diagnostic modalities, such as clinical examination and neuromuscular ultrasound, to stratify the severity of CTS. These first-line, non-invasive tools can help identify which patients are most likely to benefit from subsequent electrodiagnostic confirmation, thereby optimizing referral pathways and promoting a more efficient use of specialized diagnostic resources.

Despite extensive literature on CTS diagnosis, most studies have evaluated only one or two sonographic parameters, without incorporating dynamic assessments such as median nerve mobility. Moreover, prior work has rarely quantified the incremental value of ultrasound beyond demographic and clinical findings. There is also a lack of data from Central-Eastern Europe, where diagnostic pathways, access to electrodiagnostic testing, and epidemiologic profiles differ from Western European cohorts. In this context, the objectives of this study were to assess the correlation between ultrasound measurements and NCS-defined CTS severity and to explore clinical and demographic factors associated with CTS severity. To address these gaps, this study applied a multimodal ultrasound protocol, including static, morphologic, vascular, and dynamic parameters, to determine their independent and combined associations with NCS-defined CTS severity.

## 2. Materials and Methods

### 2.1. Study Design and Population

This cross-sectional study included all consecutive patients presenting to the Rheumatology Unit of Colentina Hospital (Bucharest, Romania) between January 2023 and March 2025, referred with a clinical diagnosis of CTS. In order to include only idiopathic cases of CTS, all patients were verified during the initial evaluation using a standardized workflow consisting of detailed medical history, physical examination, recent laboratory tests (blood and urine analyses performed within the preceding 3 months or repeated on the day of evaluation when necessary), and wrist imaging obtained at presentation. Participants were excluded from the study based on the following criteria: age under 18 years, diagnosis of acute CTS (since both ultrasound and NCS may reflect transient edema or conduction block rather than chronic structural nerve compression), history of paresis or cervical radiculopathy (confirmed by clinical examination and, when indicated, cervical imaging), presence of proximal compressive neuropathies (e.g., thoracic outlet syndrome), confirmed diabetic polyneuropathy, previous fractures or malunited callus affecting the wrist, prior CTS decompression surgery, local glucocorticoid or anesthetics injection in the carpal tunnel, or physiotherapy targeting the wrist region within the last three months, pregnancy or postpartum status up to six months, use of oral contraceptives within the preceding six months, decompensated hypothyroidism, acromegaly, chronic kidney disease requiring hemodialysis, and presence of inflammatory joint diseases such as rheumatoid arthritis, psoriatic arthritis, spondyloarthritis, systemic lupus erythematosus, gout, or other conditions associated with arthritis and tendon thickening. All study procedures (medical history, clinical interview and questionnaires, general and CTS-oriented clinical examination, ultrasound and NCS, exclusion criteria revision) were done within a 7-day span for each patient. The study was conducted in accordance with the Helsinki Declaration and was approved by the local ethics committee (number 2/19 January 2023). All patients signed informed consent before any study procedure.

### 2.2. Clinical Assessment

All participants underwent a standardized clinical assessment prior to ultrasound and NCS. The examination was performed by the same investigator (senior rheumatologist) and focused on sensory alterations in the median nerve distribution, including the median–ulnar sensory discrimination test, Tinel’s and Phalen’s signs, and the assessment of thenar atrophy and motor weakness using the lateral pinch test. Grip strength was measured with a Camry EH101 digital hand dynamometer (Zhongshan Camry Electronic Co., Ltd., Guangdong, China). Tests were performed with the arm at the side, elbow flexed at 90°, and wrist in a neutral position. Each patient performed three maximal-effort trials, with the mean value (kgf) recorded. All patients completed the Romanian version of the Boston Carpal Tunnel Questionnaire (BCTQ), recently translated and culturally validated [16].

### 2.3. Ultrasound Assessment

Ultrasonographic evaluations were conducted by a rheumatologist certified for advanced musculoskeletal ultrasound by EULAR (European Alliance of Associations for Rheumatology), who was blinded to patients’ clinical data. A MyLab Seven system (Esaote SpA, Genoa, Italy) with a 7–18 MHz linear probe was used. Patients were assessed seated, with the forearm supinated and fingers in a neutral position. Power Doppler (PD) imaging employed a 9.1 MHz frequency and 750 Hz pulse repetition rate. Each wrist was examined in both longitudinal and transverse planes, in accordance with the standardized scanning protocols outlined in the 2017 EULAR guidelines for musculoskeletal ultrasound in rheumatology [17]. Currently, there is no consensus regarding the most appropriate site or method for sonographic measurement of the median nerve in suspected cases of CTS. In this study, a multimodal ultrasonographic approach was employed, incorporating several techniques commonly cited in the literature [18,19,20,21,22] (Table 1). Three repeated measurements were obtained for each quantitative parameter and the mean was used in analyses to reduce measurement variability. Median nerve mobility was classified in a binary way, as either normal or abnormal. Abnormal mobility was defined as absent or only partial displacement of the median nerve in either the radioulnar or dorsopalmar plane during active wrist and finger motion. Frame-rate-based quantitative metrics were not part of the protocol.

### 2.4. Electrophysiological Assessment

To confirm clinical diagnoses of CTS for each hand, electrophysiological examinations were conducted using a Nihon Kohden Neuropack electromyograph (Nihon Kohden Corporation, Tokyo, Japan) with two surface cup recording electrodes and a surface electrical stimulator. Sensory and motor nerve conduction studies and F-waves of the median nerve were performed at a stimulation rate of 1 Hz. Sensory nerve action potentials (SNAPs) were recorded antidromically by stimulating the median nerve at the wrist, with recording electrodes placed on the index, middle, and ring fingers for comparative analysis, maintaining a distance of 14 cm from the wrist. Motor NCS were also performed, with compound muscle action potentials recorded following standard protocols with the active electrode positioned over the abductor pollicis brevis muscle. The stimulus duration was set at 0.2 msec across all sites. All NCS were performed by the same neurologist for consistency, who was blind to clinical and ultrasound examinations. Hand skin temperature was continuously monitored in an air-conditioned room maintained at 23–25 °C throughout the examination. The electrophysiological severity of CTS was determined according to the Bland scale [30] (from 0, normal, to 6, extremely severe). For the main analyses, these categories were collapsed into a three-point scale to ensure adequate cell counts and model stability: mild (grades 1–2), moderate (grade 3–4), and severe (grades 5–6).

### 2.5. Statistical Analysis

Nominal variables were reported as absolute frequency and percent of sub/group. Data distribution normality was assessed using descriptive statistics, normality, stem-and-leaf plots, and the Lilliefors corrected Kolmogorov–Smirnov tests. Normally distributed continuous variables are reported as “mean ± standard deviation”, while non-normally distributed continuous variables are reported as “median (interquartile range)”. Univariable associations between continuous predictors and ordinal NCS-defined CTS severity scales were examined using Spearman’s rank correlation coefficient (ρ) with corresponding 95% confidence intervals (CI), estimated via Fisher’s z-transformation. For categorical (nominal) predictors, associations with ordinal NCS severity scales were assessed using the χ^2^ test of independence. To quantify the strength of these associations, Cramer’s V was reported alongside the χ^2^
*p*-value and interpreted according to conventional thresholds (<0.1 = weak, <0.3 = moderate, ≥0.5 = strong).

Variables with statistically significant correlations in these analyses were subsequently considered for inclusion in multivariable models. To avoid collinearity, variables with very high pairwise correlations (ρ > 0.70) were not entered simultaneously into multivariable models. From each collinear group, the variable with the strongest correlation/association with NCS-defined severity was selected for inclusion into multivariable models. Because patients contributed both hands, multivariable analyses were performed at the hand level but adjusted for within-patient clustering. The models used ordinal regression by generalized estimating equations (GEE) with patient ID as the clustering variable and an exchangeable working correlation structure to obtain robust standard errors. GEE models were built sequentially. Variables were grouped into three sets based on clinical relevance rather than statistical selection: general patient characteristics, which were considered potential baseline risk factors; CTS-specific clinical findings, reflecting disease manifestation; and ultrasound parameters, representing structural and dynamic correlates of severity. Multivariable GEE models were constructed sequentially to reflect this conceptual hierarchy, not stepwise data-driven inclusion. Model 1 included demographic and core significant clinical covariates (age, sex, BMI, hypertension). Model 2 added significant clinical examination findings (vitamin supplements, motor BCTQ score, Tinel’s sign, Phalen’s sign, nocturnal symptoms, sensory loss/decrease and muscle atrophy/weakness in median nerve territory). Models 3.*x* additionally incorporated significant ultrasound parameters (median nerve maximum area, distance from the flexor retinaculum to the trapezium–pisiform line, median nerve Doppler signal, normal/abnormal median nerve mobility). Results are expressed as odds ratios (OR) with 95% confidence intervals (CI), derived by exponentiating B regression coefficients and their robust standard errors. Continuous predictors were reported as ORs per 1 standard deviation (SD) increase. In exploratory GEE models for QIC (quasi-likelihood under the independence model criterion) calculation, several clinical and ultrasound predictors were omitted due to collinearity or quasi-complete separation, resulting in unstable or identical QIC values. Therefore, QIC was not used for formal comparison of ordinal models. Instead, for each model, the log pseudolikelihood and pseudo-R^2^ (McFadden’s measure, with values in the range of 0.2–0.4 generally interpreted as good explanatory power for categorical models) were reported as indicators of relative fit. Because formal discrimination and calibration indices for ordinal GEE models are not well established, model evaluation focused on these available diagnostics. To assess the robustness of findings, sensitivity analyses were performed using a binary classification of NCS severity (severe, grades 5–6, versus non-severe, grades 1–4) and using omission tests by iteratively removing one predictor at a time from the final multivariable model. All models were re-estimated using the same GEE framework and standardized predictors.

The statistical tests were considered significant if *p* < 0.05 and were performed using IBM SPSS Statistics version 26.0 for Windows (IBM Corp., Armonk, NY, USA), Stata version 12 (StataCorp, College Station, TX, USA) and R version 4.5.1 (R Foundation for Statistical Computing, Vienna, Austria).

## 3. Results

### 3.1. Demographic, Clinical, Ultrasound and NCS Characteristics

The study cohort included 100 patients (Table 2) had a mean age of 58.5 ± 9.8 years, with a predominance of female participants (93.0%), most of whom reported menopause at a mean age of 47.8 ± 5.6 years. The majority of participants resided in urban areas (62.0%). Educational attainment was distributed as follows: 10.0% had completed only secondary school, 64.0% high school, and 26.0% college. More than half of the cohort were professionally active (57.0%). Current smoking was reported by 18.0% of participants, while 37.0% reported having smoked before (ever smoking); frequent alcohol consumption was noted in 48.0%. The mean body mass index (BMI) was 29.2 ± 5.0 kg/m^2^, and arterial hypertension was present in 57.0% of patients. Raynaud’s phenomenon was reported by 3.0% of the cohort.

In the hand sample (*n* = 193; Table 2), the mean age at CTS onset was 54 ± 11 years, with diagnosis occurring at a mean age of 57 ± 10 years. Nearly half of the patients (48.2%) reported use of vitamin B supplements. The majority experienced nocturnal symptoms (75.1%), symptoms worsening in an elevated hand position (77.2%), or with repetitive hand movements (66.3%). Symptom alleviation through shaking or flicking of the hand was reported by 67.9% of patients, while changes in hand position were associated with symptom relief in 71.5% of cases. On clinical examination, Tinel’s sign was present in 39.9% and Phalen’s sign in 64.8%. Morning hand stiffness was reported by 74.1%. Sensory loss in the median nerve territory was observed in 36.8%, thenar muscle weakness/atrophy in 20.7%, and dry skin in 11.4%. The mean BCTQ scores were 32 ± 10 for the sensory domain and 21 ± 8 for the motor domain. Mean grip strength was 19.4 ± 5.8 kgf.

Ultrasound evaluation of the median nerve in CTS patients showed a mean cross-sectional area (CSA) of 13.8 ± 3.8 mm^2^ at the carpal tunnel inlet (iCSA), 9.8 ± 2.7 mm^2^ at the outlet (oCSA), and 6.8 ± 1.4 mm^2^ at the distal forearm (pCSA), with a maximal CSA (mCSA) of 15.3 ± 4.2 mm^2^ (Table 1). The corresponding flattening ratios were 3.1 ± 0.8 at the inlet (iFR) and 2.9 ± 0.7 at the outlet (oFR). The mean wrist-to-forearm CSA ratio was 2.1 ± 0.6. Power Doppler signal within the median nerve was observed in 68.4% of cases, while reduced echogenicity was present in 96.9% and abnormal mobility in 60.1%. Flexor retinaculum bowing (FRB) averaged 3.8 ± 1.2 mm. Inlet median nerve tapering was detected in 90.7% of patients, outlet tapering in 20.2%, and anatomical variants (such as bifid nerve or persistent median artery) in 11.9%.

NCS revealed in the hand sample (*n* = 193) a mean sensory nerve action potential (SNAP) amplitude of 15.9 ± 11.0 µV and a sensory conduction velocity (SCV) of 37.5 ± 11.0 m/s. The mean distal motor latency (DML) was 5.4 ± 1.9 ms, with a compound muscle action potential (CMAP) amplitude of 7.2 ± 2.8 mV and a motor conduction velocity (MCV) of 54.6 ± 7.3 m/s. Mean motor F-wave latency measured 28.6 ± 4.2 ms. Based on electrodiagnostic grading, the distribution across the 0–6 scale was: grade 1, 5.7%; grade 2, 23.3%; grade 3, 42.5%; grade 4, 4.1%; grade 5, 17.1%; and grade 6, 7.3% (Figure 1). When collapsed into the 3-point severity scale, 29.0% of hands were classified as mild (grades 1–2), 46.6% as moderate (grade 3–4), and 24.4% as severe (grades 5–6).

### 3.2. Univariate Analysis

Several demographic (age at study entry, age at CTS onset, age at CTS diagnosis, sex, BMI, hypertension), clinical (sensory and motor BTCQ, symptoms during sleep, Tinel’s sign, Phalen’s sign, decrease/loss of sensitivity in median nerve territory, muscle weakness/atrophy in median nerve territory, vitamin supplements), and ultrasound parameters (maximal cross-sectional area of median nerve—mCSA; CSA at the carpal tunnel inlet—iCSA; perpendicular distance from the flexor retinaculum to the trapezium–pisiform line—FRB, median nerve Doppler signal, median nerve mobility) demonstrated significant correlations with both the extended NCS-defined CTS severity scale and the collapsed 1–3 scale (Table 3). The rest of the clinical CTS characteristics (worsening symptoms in sloping position of hand, worsening symptoms with repetitive hand movement, improving symptoms by hand shaking/flicking, improving symptoms at changing hand position, morning hand stiffness, Raynaud phenomena, dry skin in median nerve territory, and grip strength) did not exhibit significant correlations/associations with NCS-defined CTS severity.

### 3.3. Multivariate Analysis

In the base GEE model including only demographic and general clinical factors (model 1: age, sex, BMI, hypertension), none were independently associated with NCS severity on the 3-point scale (Table 4). When CTS-specific symptoms and signs were added (model 2), male sex, nocturnal symptoms, sensory loss in the median nerve territory, and thenar weakness/atrophy emerged as significant predictors of higher electrodiagnostic severity, while other clinical variables were not significant. Incorporation of ultrasound parameters produced more variable results across specifications (models 3.1–3.7). Maximum median nerve area was the most consistent imaging correlate of NCS severity, retaining significance in several models (models 3.1, 3.3, 3.6), whereas distance from the flexor retinaculum to the trapezium–pisiform line and median nerve Doppler signal showed inconsistent or non-significant associations. Median nerve mobility became significant in the fully adjusted model including all ultrasound variables (model 3.7). Overall, the most robust independent predictors across models were sex, nocturnal symptoms, sensory loss, thenar weakness/atrophy, and maximum median nerve area, indicating that a combined clinical and sonographic assessment best reflected NCS-defined CTS severity in this cohort. Model fit improved progressively with the addition of CTS-specific clinical and ultrasound predictors. The core clinical model (model 1) showed poor explanatory power (log pseudolikelihood −195, pseudo-R^2^ = 0.04). Adding CTS clinical signs and symptoms (model 2) substantially improved fit (−169; 0.17). Inclusion of ultrasound parameters led to further incremental gains, with log pseudolikelihood values rising from −162 (model 3.1) to −146 (model 3.7) and pseudo-R^2^ values increasing from 0.19 to 0.25. The best-fitting specification was model 3.7, which incorporated both clinical and ultrasound measures, indicating that ultrasound adds complementary information beyond standard clinical assessment. Sensitivity analyses confirmed the robustness of the primary findings. Alternative severity groupings yielded similar significant predictors and comparable effect sizes. Omission tests did not materially change any remaining coefficients, indicating that no single variable disproportionately influenced the model.

## 4. Discussion

The present study achieved its objectives by identifying several ultrasound parameters, particularly the maximum cross-sectional area of the median nerve and median nerve mobility, which were significantly associated with electrophysiological severity. In addition, clinical factors such as nocturnal symptoms, sensory loss, and weakness/atrophy in the median nerve territory, together with male sex, were independently linked to more severe CTS. In contrast, general demographic variables such as age, BMI, and hypertension did not remain significant in multivariable models. These results confirm that ultrasound measurements provide complementary information to NCS, while clinical features continue to play an important role in characterizing CTS severity. Clinically, these results indicate that ultrasound can provide valuable complementary information to NCS in the evaluation of CTS severity, particularly through assessment of median nerve cross-sectional area and mobility. The identification of clinical features such as nocturnal symptoms, sensory loss, and thenar weakness as independent correlates of severity further underscores the importance of thorough bedside assessment. Together, these parameters may support earlier recognition of patients with more advanced disease and expand diagnostic options in settings where NCS is less accessible. A novel contribution of this study is the identification of a reproducible clinical sonographic profile associated with electrophysiological severity: male sex, nocturnal symptoms, sensory loss, thenar weakness, increased maximal CSA, and impaired nerve mobility. This composite profile may support more effective clinical triage, inform decisions on expedited NCS referral, and guide early consideration of decompression in severe cases. Also, by quantifying the incremental value of ultrasound using a sequential GEE strategy, we demonstrate that sonography meaningfully enhances severity assessment beyond clinical findings.

Our results, showing that a larger median nerve CSA and abnormal nerve mobility are independently associated with the electrophysiological severity of CTS, align with accumulating evidence supporting their prognostic value. Several ultrasonographic studies have reported that median nerve CSA correlates positively with NCS severity.

For example, Panicker and Iype reported that only the median nerve CSA demonstrated a significant correlation with electrodiagnostic severity (r = 0.545), whereas flattening ratio and wrist–forearm ratio showed no significant associations [31]. Likewise, Potužník et al. found strong associations between CSA and both sensory conduction velocity (τ = −0.516) and distal motor latency (τ = 0.587), with excellent diagnostic accuracy (AUC = 0.93) and an optimal threshold of ≥12 mm^2^ for detecting moderate–severe CTS [32]. More recently, studies from 2024 further demonstrated stepwise increases in CSA, delta-CSA, and wrist-forearm ratio across electrophysiologic severity categories, with the highest values in severe CTS and statistically significant differences between severity groups (all *p* < 0.001) [33,34]. Recent evidence highlights that both longitudinal and transverse nerve dynamics deteriorate progressively with increasing CTS severity. Rossmann et al. demonstrated that longitudinal gliding is markedly reduced in CTS, decreasing from 5–6 mm in healthy controls to approximately 2 mm in weakly positive NCS cases, and becoming nearly absent in moderate and severe disease; although effective in distinguishing patients from controls, it did not reliably discriminate between electrophysiologic severity subgroups [35]. Similarly, Thomas et al. reported significant reductions in transverse nerve mobility, with movement falling from 14% in controls to 6.8% in cases overall and showing a clear stepwise decline across NCS-defined severity grades [36]. Complementing these findings, Lo et al. [37] found that reduced nerve movement within the carpal tunnel is inversely associated with symptom severity. Collectively, these data suggest that dynamic nerve mobility, whether longitudinal or transvers, provides meaningful physiological insight into CTS progression and may enhance severity stratification alongside conventional NCS measures. These findings also reinforce the clinical value of ultrasound assessment beyond static nerve size: combining CSA and mobility measurements enhances the ability to identify more severe cases of CTS, potentially streamlining risk stratification and intervention decisions. From a clinical perspective, these findings have several practical implications. First, ultrasound offers a rapid, non-invasive method to approximate disease severity in everyday practice, which may be particularly useful where NCS is not readily available. Second, the integration of dynamic mobility assessment into standard ultrasound protocols may improve the ability to identify patients at higher risk of progression or those more likely to require surgical decompression. Third, combining static (CSA) and dynamic (mobility) ultrasound parameters could enhance diagnostic accuracy, streamline patient triage, and reduce reliance on more invasive or uncomfortable electrophysiological testing, especially in patients reluctant to undergo NCS. Ultimately, the incorporation of these measurements may support earlier and more individualized intervention strategies in CTS management.

The finding that male sex is independently associated with greater electrophysiological severity of CTS aligns with previous observations that men may tolerate symptoms longer or present later in the disease course, leading to higher NCS grades at diagnosis. In a large multicenter Italian study, men reported less discomfort and better hand function than female participants despite showing more severe nerve impairment on neurophysiologic testing [38]. Also, men present with a higher burden of comorbidities [39], which can expose them to higher NCS degrees of CTS severity. Conversely, female participants report their CTS symptoms earlier and more frequently [40], and they may be more vulnerable to work-related median nerve dysfunction, potentially because of occupational factors, resulting in more severe CTS presentations, particularly in manual labor contexts [41]. This work-related association seems to be related to the frequency of female participants engaging them, rather than the manual labor itself, which determines similar rates of CTS among sexes [42]. Anatomical factors may also contribute to this disparity: magnetic resonance imaging data indicate that female participants have a constitutionally smaller carpal tunnel cross-sectional area relative to hand size compared with men, a difference that could predispose to CTS development [43]. These findings suggest that sex-related differences in symptom perception or health-seeking behavior may contribute to clinically meaningful disparities in CTS severity at diagnosis. Clinicians should therefore consider a lower threshold for diagnostic testing in female participants with early symptoms, while maintaining a high index of suspicion for severe disease in men who seek care only after prolonged or progressive complaints.

Nocturnal symptoms (numbness, tingling, and pain that awaken patients at night) are not only hallmark features of CTS, but also important clinical indicators of disease severity. Prior work has shown that symptoms worsen during sleep due to wrist malposition and increased pressure within the carpal tunnel, which acutely exacerbate median nerve compression [44,45]. Clinically, symptoms that disturb sleep have been repeatedly emphasized as highly suggestive of CTS and often prompt referral for diagnostic studies. Mechanistically, nocturnal symptom exacerbation reflects elevated intraneural ischemia and mechanosensitive fiber activation—pathophysiological features associated with more advanced neuronal injury [46]. Our findings that nocturnal symptoms are independently associated with higher electrophysiological severity support their value as a prognostic marker. Thus, when patients report night-time hand symptoms, clinicians should maintain a lower threshold for prompt and comprehensive evaluation, as this may signal more severe underlying nerve compromise.

In addition to electrophysiological and ultrasound markers, we evaluated functional and patient-reported measures such as grip strength and the Boston Carpal Tunnel Questionnaire (BCTQ). While BCTQ scores capture patient-perceived symptom burden and function and have strong psychometric support, their relationship with electrophysiological severity is generally modest and not consistently discriminatory across NCS grades [47,48]. By contrast, our ultrasound parameters, particularly maximal CSA and mobility, tracked more closely with NCS severity. Grip strength also showed only limited association with NCS grades, consistent with prior work noting weak correspondence between clinical outcomes and neurophysiologic measures [49]. Taken together, these comparisons suggest that although clinical scores and functional tests remain useful for documenting patient experience and disability, ultrasound markers may better approximate electrophysiological severity and thus contribute more directly to severity grading and treatment planning.

Overall, our results support the view that ultrasound should be considered a valuable adjunct rather than a replacement for NCS in the evaluation of CTS severity. When both modalities are available, ultrasound findings, particularly maximal median nerve CSA and abnormal mobility, can reinforce electrophysiological grading, improve diagnostic confidence, and help prioritize patients for timely intervention. In clinical practice, concordant evidence from both tests may accelerate surgical referral in severe cases or provide reassurance for conservative management in milder cases. In resource-limited settings where NCS is not readily accessible, ultrasound offers a rapid, non-invasive, and patient-friendly means of approximating CTS severity, thereby guiding treatment decisions without delay. While NCS remains indispensable for definitive diagnosis in complex or atypical presentations, the integration of ultrasound into routine pathways has the potential to reduce reliance on invasive, time-consuming, and costly studies, particularly in patients with typical clinical features and characteristic sonographic findings. Future studies should build on our findings by incorporating longitudinal follow-up to determine whether ultrasound markers such as maximal CSA and nerve mobility not only correlate with but also predict clinical progression and treatment response. Multicenter studies with standardized ultrasound protocols are needed to confirm reproducibility across operators and machines, and to establish clinically meaningful cut-offs for severity grading. Integration of patient-reported outcomes with imaging and electrophysiological data could further refine risk stratification models. Finally, comparative effectiveness research is warranted to evaluate whether incorporating ultrasound into routine diagnostic pathways can reduce the need for invasive NCS, shorten time to treatment, and improve patient-centered outcomes.

Several methodological aspects should be considered when interpreting these findings. First, the cross-sectional design precludes causal inference regarding the relationship between ultrasound features and CTS severity. Although NCS was taken as the reference standard, its limitations in detecting all clinically evident cases must be acknowledged. The study benefited from consecutive patient inclusion and the use of GEE models to account for the correlation between hands, but the moderate sample size and single-center setting may introduce spectrum bias and restrict generalizability (multicenter validation is needed). Ultrasound measurements were operator-dependent, and some clinical variables relied on patient self-report. A further limitation of our study is the absence of formal intra- and inter-rater reliability testing for key ultrasound parameters, although previous studies have demonstrated good to excellent reliability for most median nerve and wrist measurements [50,51]. Finally, QIC could not be calculated for ordinal GEE models in order to compare their goodness of fit. Additional limitations include the restriction of analyses to NCS-confirmed CTS hands, which may limit applicability to patients with clinically defined but NCS-negative disease. Recruitment from a specialized clinic may introduce spectrum bias toward more severe cases. No longitudinal data were available to determine whether ultrasound or clinical predictors forecast treatment response. Moreover, only a limited set of comorbidities were examined/recorded.

## 5. Conclusions

This study demonstrates that specific ultrasound parameters, particularly maximal median nerve CSA and abnormal nerve mobility, are significantly associated with electrophysiological severity of CTS. In addition, clinical features such as nocturnal symptoms, sensory loss, and thenar weakness/atrophy, together with male sex, were independent correlates of severity, whereas general demographic factors showed no predictive value. These findings highlight the complementary role of ultrasound alongside NCS in the assessment of CTS severity and suggest that integrating structural and dynamic sonographic markers with clinical evaluation can improve risk stratification and guide timely intervention. Further research should validate standardized ultrasound cut-offs and explore their utility in predicting treatment outcomes.

## Figures and Tables

**Figure 1 diagnostics-15-02985-f001:**
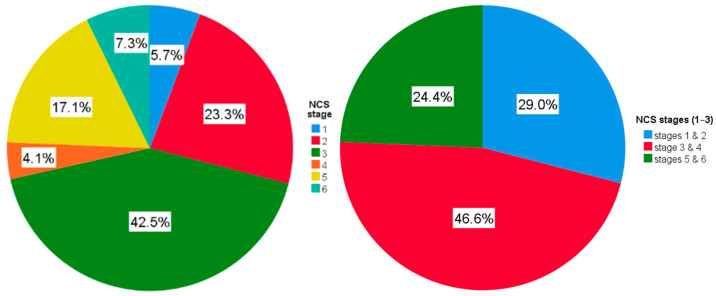
NCS stages of CTS on the 1–6 scale (**left** pane) and on the 1–3 scale (**right** pane). Abbreviations: CTS—carpal tunnel syndrome; NCS—nerve conduction study.

**Table 1 diagnostics-15-02985-t001:** Definitions and values of CTS ultrasound parameters.

Parameter Name	Definition of Parameter	Cut-Off Value	In Sample (*n* = 193)
iCSA	Measured at the pisiform *	10 mm^2^ [23]	13.8 ± 3.8
oCSA	Measured at the hamate’s hook *	10 mm^2^ [24]	9.8 ± 2.7
mCSA	Largest CSA measured, irrespective of bony landmarks *	10 mm^2^ [25]	15.3 ± 4.2
pCSA	CSA at the proximal third of the pronator quadratus *	-	6.8 ± 1.4
iFR	Ratio of long to short axis of the median nerve at the tunnel inlet	>2.5 [26]	3.1 ± 0.8
oFR	Ratio of long to short axis of the median nerve at the tunnel outlet	>2.6 [27]	2.9 ± 0.7
CSA ratio	Ratio of CSA at the carpal tunnel inlet to CSA at the proximal third of the pronator quadratus	1.4 [28]	2.1 ± 0.6
PD signal	Detection of intraneural vascular signal within the MN	absent/ present	68.4%
MN echogenicity	Evaluation of internal pattern of the median nerve (e.g., normal, hypoechoic, hyperechoic)	normal **/ abnormal	96.9%
MN mobility	Assessment of nerve displacement during wrist and finger movement	normal ***/ abnormal [29]	60.1%
FRB	Perpendicular distance from the flexor retinaculum to the trapezium–pisiform line	2.5 [24]	3.8 ± 1.2
inlet median nerve tapering	Visual/qualitative alteration in median nerve diameter at the inlet in longitudinal view	absent/ present	90.7%
outlet median nerve tapering	Visual/qualitative alteration of the MN’s caliber at the carpal tunnel outlet in longitudinal view	absent/ present	20.2%
anatomical variants	Detection of bifid MN, persistent median artery, or accessory tendons/muscles	absent/ present	11.9%

Notes: * by tracing a continuous contour along the inner margin of the nerve’s hyperechogenic boundary; ** Fascicular pattern; *** Transverse and deep gliding of the MN. Abbreviations: CTS—carpal tunnel syndrome; FRB—flexor retinaculum bowing; i/oCSA—cross-sectional area at the carpal tunnel inlet/outlet; i/oFR—flattening ratio at the carpal tunnel inlet/outlet; mCSA—maximal CSA; median nerve—median nerve; pCSA—CSA at the distal forearm; PD—power Doppler.

**Table 2 diagnostics-15-02985-t002:** General and CTS characteristics.

Patients (*n* = 100)		Hands (*n* = 193)	
Age (y)	58.5 ± 9.8	Age at CTS onset (years)	54 ± 11
Female	93.0%	Age at CTS diagnosis (years)	57 ± 10
Menopause age (y)	47.8 ± 5.6	Vitamin B supplements	48.2%
Urban dwelling	62.0%	CTS symptoms during sleep	75.1%
Secondary school	10.0%	Worsening symptoms in elevated position of hand	77.2%
High school	64.0%	Worsening symptoms with repetitive hand movement	66.3%
College	26.0%	Worsening symptoms upon changing hand position	71.5%
Professional active	57.0%	Improving symptoms with hand shaking/flicking	67.9%
Smoking, current	18.0%	Tinel’s sign	39.9%
Smoking, ever	37.0%	Phalen’s sign	64.8%
Alcohol, frequent	48.0%	Morning hand stiffness	74.1%
BMI (kg/m^2^)	29.2 ± 5.0	Decrease/loss of sensitivity in median nerve territory	36.8%
Arterial hypertension	57.0%	Muscle weakness/atrophy in median nerve territory	20.7%
Raynaud phenomena	3.0%	Dry skin in median nerve territory	11.4%
		BCTQ (sensory)	32 ± 10
		BCTQ (motor)	21 ± 8
		Grip strength (kgf)	19.4 ± 5.8

Abbreviations: BMI—body mass index; BTCQ—Boston Carpal Tunnel Questionnaire; CTS—carpal tunnel syndrome; y—years.

**Table 3 diagnostics-15-02985-t003:** Significant correlations and associations of clinical and ultrasound variables with NCS-defined CTS severity * (*n* = 193).

Clinical and Ultrasound Variables	*ρ*	*p*	95% CI
Age at study entry	0.229	0.003	0.076–0.383
Body mass index	0.169	0.027	0.028–0.308
Age at CTS onset	0.197	0.010	0.049–0.356
Age at CTS diagnosis	0.188	0.014	0.040–0.351
BCTQ (sensitive)	0.252	0.001	0.111–0.389
BCTQ (motor)	0.365	0.000	0.235–0.487
mCSA	0.298	0.000	0.154–0.426
iCSA	0.240	0.002	0.086–0.372
FRB	0.344	0.000	0.184–0.478
CSA ratio	0.196	0.010	0.043–0.330
	*χ* ^2^	*p*	Cramer’s *V*
Sex	11.9	0.037	0.249
Arterial hypertension	12.5	0.029	0.256
Vitamin supplements	19.2	0.002	0.317
CTS symptoms during sleep	12.8	0.025	0.259
Tinel’s sign	12.4	0.030	0.254
Phalen’s sign	10.9	0.048	0.239
Decrease/loss of sensitivity in median nerve territory	26.1	0.000	0.370
Muscle weakness/atrophy in median nerve territory	54.5	0.000	0.534
MN Doppler signal	25.1	0.000	0.362
Partial/no median nerve mobility	23.9	0.000	0.353

Note: * collapsed from the Bland scale to mild (grades 1–2), moderate (grade 3), and severe (grades 4–6). Abbreviations: BCTQ—Boston Carpal Tunnel Syndrome Questionnaire; CSA—cross sectional area; CTS—carpal tunnel syndrome; FRB—perpendicular distance from the flexor retinaculum to the trapezium–pisiform line; iCSA—CSA at the carpal tunnel inlet; mCSA—maximal CSA; median nerve—median nerve; NCS—nerve conduction study.

**Table 4 diagnostics-15-02985-t004:** Multivariable GEE models predicting NCS-defined CTS severity (1–3 scale).

Predictor (OR, 95% CI)/Model	1	2	3.1	3.2	3.3	3.4	3.5	3.6	3.7
Age at study entry (y)	1.03	0.99	1.01	1	1.03	0.99	1	1.02	1.03
	0.99–1.08	0.93–1.04	0.95–1.06	0.94–1.06	0.97–1.09	0.93–1.05	0.93–1.06	0.96–1.09	0.97–1.10
Sex (male)	2.26	**6.16** *	**9.99** *	5.2	**9.12** *	**6.20** *	5.16	**8.59** *	3.88
	0.58–8.85	1.34–28.2	1.13–88.5	0.96–28.0	1.42–58.6	1.37–28.1	0.92–29.1	1.46–50.4	0.59–25.3
Body mass index (kg/m^2^)	1.04	1	1	1.01	0.97	1.01	1	0.98	1.01
	0.97–1.12	0.88–1.14	0.89–1.13	0.88–1.14	0.85–1.11	0.89–1.13	0.89–1.13	0.87–1.12	0.90–1.13
Arterial hypertension (no)	0.78	0.6	0.52	0.7	0.62	0.59	0.64	0.67	1.1
	0.34–1.80	0.20–1.79	0.18–1.52	0.23–2.15	0.19–2.01	0.20–1.77	0.22–1.87	0.20–2.22	0.40–3.02
Supplements (no)	-	1.46	1.47	1.64	1.68	1.48	1.8	1.5	1.22
		0.56–3.77	0.54–3.95	0.64–4.19	0.66–4.31	0.57–3.87	0.70–4.61	0.58–3.86	0.53–2.83
Symptoms in sleep (no)	-	**0.38** *	**0.42** *	**0.36** *	0.37	**0.38** *	**0.36** *	**0.37** *	**0.33** *
		0.19–0.77	0.18–0.96	0.15–0.85	0.12–1.14	0.18–0.81	0.15–0.85	0.14–0.99	0.11–0.98
Tinel’s sign (no)	-	0.8	1.07	0.77	1.12	0.83	0.81	1.14	0.7
		0.46–1.42	0.49–2.30	0.37–1.60	0.44–2.88	0.44–1.58	0.38–1.72	0.47–2.73	0.25–1.98
Phalen’s sign (no)	-	1.34	1.48	1.45	1.62	1.39	1.47	1.56	1.7
		0.55–2.29	0.27–1.46	0.44–2.30	0.28–1.44	0.52–2.25	0.40–2.11	0.32–1.26	0.55–1.80
Decrease/loss of sensitivity (no)	-	**0.43** *	0.67	**0.44** *	0.49	**0.42** *	**0.44** *	**0.44** *	0.61
		0.23–0.82	0.23–1.89	0.22–0.86	0.21–1.11	0.23–0.79	0.23–0.82	0.21–0.93	0.30–1.24
Muscle weakness/atrophy (no)	-	**0.26** *	**0.28** *	**0.32** *	0.37	**0.24** *	**0.30** *	0.39	0.35
		0.10–0.64	0.10–0.83	0.12–0.89	0.12–1.15	0.10–0.60	0.10–0.90	0.14–1.10	0.12–1.01
Motor BCTQ	-	1.06	1.02	1.05	1.03	1.06	1.05	1.03	1.01
		0.98–1.15	0.96–1.09	0.96–1.16	0.96–1.11	0.98–1.14	0.95–1.15	0.96–1.11	0.90–1.14
MN maximum area (mm^2^)	-	-	1.12	-	**1.15** *	-	-	**1.16** *	1.09
			0.98–1.28		1.02–1.29			1.03–1.30	0.96–1.24
Retinaculum-SP line distance (mm)	-	-	-	1.15	1.23	-	1.14	1.2	1.22
				0.83–1.60	0.93–1.64		0.82–1.59	0.85–1.68	0.73–2.05
MN Doppler signal (no)	-	-	-	-	-	1.13	1.01	0.91	1.06
						0.55–2.30	0.34–2.99	0.42–2.00	0.19–6.02
Partial/no MN mobility (no)	-	-	-	-	-	-	-	-	**0.36** *
									0.17–0.74
Log pseudolikelihood	−195	−169	−162	−158	−154	−164	−152	−150	−146
pseudo-R^2^	0.04	0.17	0.19	0.18	0.2	0.19	0.21	0.22	0.25

Notes: significant predictors (*p* < 0.05) are bold and marked with *. Abbreviations: CI—confidence interval; CTS—carpal tunnel syndrome; MN—median nerve; NCS—nerve conduction studies; OR—odds ratio.

## Data Availability

Dataset available on request from the authors.

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
