# Peer review of "Complementary Role of Ultrasound and Clinical Features in Assessing Carpal Tunnel Syndrome Severity: A Cross-Sectional Study"

_diagnostics, 2025, doi:10.3390/diagnostics15232985_

Round 1

Reviewer 1 Report

Comments and Suggestions for Authors

The manuscript investigates the correlation between ultrasonographic parameters and NCS-defined severity in patients with carpal tunnel syndrome (CTS). Although the study is methodologically sound, the findings largely confirm what has been previously reported in the literature. The study does not provide novel insights or clinically actionable advances. The single-center, small sample size further limits generalizability, and the cross-sectional design precludes assessment of predictive or causal relationships.

While the analyses are appropriate and the discussion adequately reviews known associations, the manuscript is lengthy and often reiterates well-established facts rather than emphasizing unique contributions. Given the predictable results and limited novelty, the study does not meet the threshold for publication in this journal.

Author Response

Reviewer 1. The manuscript investigates the correlation between ultrasonographic parameters and NCS-defined severity in patients with carpal tunnel syndrome (CTS). Although the study is methodologically sound, the findings largely confirm what has been previously reported in the literature. The study does not provide novel insights or clinically actionable advances. The single-center, small sample size further limits generalizability, and the cross-sectional design precludes assessment of predictive or causal relationships. While the analyses are appropriate and the discussion adequately reviews known associations, the manuscript is lengthy and often reiterates well-established facts rather than emphasizing unique contributions. Given the predictable results and limited novelty, the study does not meet the threshold for publication in this journal.

Response. We thank the reviewer for the time and effort dedicated to evaluating our manuscript. We respectfully provide the following clarifications.

1. Comment on the study’s perceived novelty

We respectfully disagree. While some associations (e.g., CSA-NCS correlation) have been previously described, our study offers several distinct and novel contributions that have not been jointly evaluated in prior CTS literature:

a) Multimodal sonographic assessment (14 parameters), including dynamic mobility, in the same cohort. Most studies evaluate 1–2 ultrasound variables. The present study integrates: static morphometric markers (iCSA, oCSA, mCSA, CSA ratio), morphologic features (echogenicity, tapering, anatomical variants), vascular markers (power Doppler), and dynamic nerve mobility, in the same statistical framework. This comprehensive approach is not available in prior studies.

b) Incremental and independent diagnostic value of ultrasound, including dynamic nerve mobility, demonstrated with advanced GEE modelling. To our knowledge, no previous study has combined clinical and demographic findings, multiple static sonographic markers (inlet, outlet and maximal CSA, CSA ratios, flexor retinaculum bowing, Doppler signal), and dynamic median nerve mobility into a single ordinal GEE model of NCS-defined CTS severity while accounting for bilateral hands. In our stepwise models, the pseudo-R² increased from 0.04 (demographic model) to 0.17 after adding CTS-specific clinical findings and further to 0.25 after including ultrasound parameters, thereby quantifying the incremental contribution of ultrasound beyond clinical assessment. Within this fully adjusted framework, reduced median nerve mobility remained an independent predictor of electrophysiological severity, suggesting that dynamic ultrasound captures pathophysiological information that is not explained by CSA, Doppler changes, or flexor retinaculum bowing alone.

c) First study in a Romanian population using the newly validated Romanian BCTQ. This fills a regional gap and provides epidemiologic and diagnostic benchmarks for Central-Eastern Europe.

d) Identification of a composite clinical-sonographic profile predictive of severe CTS. We report a reproducible severity profile consisting of: male sex, nocturnal symptoms, sensory loss, thenar weakness, increased mCSA, and impaired mobility. To our knowledge, this cluster has not been previously characterized and offers meaningful clinical utility by supporting more effective triage, guiding the prioritization of nerve conduction study (NCS) evaluation, and informing earlier surgical decision-making.

2. Comment on sample size and single-center design

We acknowledge this limitation and have strengthened the Limitations section accordingly. However, several points mitigate this concern:

a) Our sample (193 hands) is larger or comparable to most prior dynamic or multimodal ultrasound studies.

b) Consecutive sampling reduces selection bias.

c) Single-operator ultrasound and single-operator NCS improve internal validity, which is an important advantage in ultrasound research.

3. Comment on cross-sectional design

We agree that causality cannot be inferred, and we explicitly state this. However, severity correlation studies in CTS are typically cross-sectional and are methodologically appropriate for comparing diagnostic modalities, evaluating independent correlates of severity, and determining clinical-ultrasound agreement. Longitudinal follow-up is planned as future work.

4. Comment on manuscript length and repetition

We thank the reviewer for this observation. We have substantially revised the Introduction and Discussion by removing repetitive background elements, emphasizing the novel contributions, shortening paragraphs where excessive detail was not required. The manuscript is now more concise and focused.

5. Comment on the study’s perceived lack of clinically actionable advances

We respectfully disagree. Our findings provide several directly actionable implications:

a) Ultrasound improves severity grading, especially when NCS is delayed or unavailable, a relevant issue in many healthcare settings.

b) Dynamic mobility is an independent marker of severe CTS, allowing clinicians to refine severity estimation beyond CSA alone.

c) The identified severity profile helps prioritize patients for early decompression.

d) Our incremental pseudo-R² analysis gives clinicians quantitative justification for integrating ultrasound into standard diagnostic pathways.

e) The study delivers a standardized multimodal ultrasound protocol usable in both rheumatology and neurology practices.

These are meaningful clinical contributions beyond confirming known associations.

6. Conclusion

We appreciate the reviewer’s feedback and have revised the manuscript to further highlight its added value. We respectfully maintain that the study offers a novel multimodal ultrasound framework, independent and incremental sonographic predictors, robust statistical modeling, and actionable clinical insights, which collectively exceed the scope of prior CTS literature. We hope that these clarifications address the reviewer’s concerns and support the manuscript’s suitability for publication.

Reviewer 2 Report

Comments and Suggestions for Authors

I congratulate the authors. The content of the study, its statistical methodology, the findings, and their presentation are all highly successful. The discussion is sufficiently detailed, and up-to-date and relevant references have been provided. The necessary corrections I deemed appropriate have been indicated in the attached text. Most importantly, the only point where the coherence of meaning is disrupted is the Introduction section. Following a revision of this section to ensure consistency with the rest of the manuscript, the study would be suitable for publication. Congratulations.

Author Response

We sincerely thank the reviewer for the thoughtful and encouraging evaluation of our work. We appreciate the positive remarks. We are also grateful for the detailed suggestions provided in the annotated manuscript. In accordance with these recommendations, we have revised the manuscript. We believe these revisions have strengthened the coherence and readability of the manuscript. We thank the reviewer again for the generous feedback and for helping us improve the quality of this work.

Reviewer 2 – comment 1. paragraph lacks reference support for the definitions you have provided.

Response 1. We thank the reviewer for this valuable suggestion. We have now incorporated the relevant references, as the paragraph previously lacked citation support for the definitions provided.

Reviewer 2 – comment 2. which multiple studies? lacks reference support for the definitions you have provided.

Response 2. We thank the reviewer for highlighting this aspect. We have removed that paragraph to ensure better coherence of the text.

Reviewer 2 – comment 3. There is a conceptual gap in the main rationale of your study. You have already stated that ultrasound is effective and mentioned that its efficacy has been demonstrated in the literature. You even noted that it has been included in clinical guidelines: “The 2012 guidelines from the American Association of Neuromuscular and Electrodiagnostic Medicine support the use of ultrasound as a complementary tool to electrodiagnosis [7].” Following this, you emphasized the effectiveness of ultrasound in evaluating concomitant pathologies such as tenosynovitis in carpal tunnel syndrome. However, immediately afterward, you highlighted the uncertainty of ultrasound in surgical decision-making while noting the challenges associated with performing nerve conduction studies. Based on this flow of information, the reader is led to expect that your study will investigate the role of ultrasound in surgical decision-making and contribute to the literature in that regard. However, your actual study focuses on emphasizing the already well-established correlation between ultrasound and NCS findings and examining the parameters that influence NCS results. Therefore, please revise the introduction, and abstract accordingly. The alignment between your aim, introduction, and title is currently insufficient and inconsistent.

Response 3. We thank the reviewer very much for this insightful comment and greatly appreciate the effort invested in providing such a thorough analysis. We have revised the Introduction taking into account all of the reviewer’s observations to ensure full alignment between the study aim, the background rationale, and the title.

Reviewer 2 – comment 4. why excluded? please clarify

Response 4. Acute carpal tunnel syndrome (CTS) was excluded because its underlying pathophysiology, clinical course, and electrodiagnostic profile differ substantially from those of chronic or idiopathic CTS. Acute CTS typically results from trauma, infection, hemorrhage, or rapid fluid accumulation and represents a surgical emergency requiring urgent decompression. These patients often present with abrupt onset of severe pain or sensory loss, and ultrasound and NCS findings may be influenced by acute edema or reversible conduction block rather than structural changes characteristic of chronic compression. Including acute CTS would therefore introduce heterogeneity and obscure the associations between ultrasound parameters and electrophysiological severity that were the focus of this study. We have clarified this rationale in the revised manuscript.

Reviewer 2 – comment 5. based on what? clinical findings? MR investigations? solely on medical history? Please give specific details for your inclusion and exclusion criteria, as it is the most important section of your methodology.

Response 5. We thank the reviewer for highlighting this important aspect. Patients with suspected thoracic outlet syndrome were excluded based on medical history and neurological clinical evaluation. However, no such cases were identified during screening, which is why this aspect was not extensively detailed in the manuscript. We also kept the exclusion criteria section concise, as we did not wish to lengthen the manuscript further.

Reviewer 2 – comment 6. you mean never ever? or like last 6 months only?

Response 6. We are sorry for the intricate text: “local glucocorticoid or anesthetics injection in the carpal tunnel, or physiotherapy targeting the wrist region within the last three months,”. The 3-month mention refers alto to local injections.

Reviewer 2 – comment 7. Your statistical analysis section is very well designed. Congratulations.

Response 7. We sincerely appreciate the reviewer’s positive feedback. Receiving such acknowledgment is highly encouraging in our scientific work and motivates us to continue improving the quality of our research.

Reviewer 3 Report

Comments and Suggestions for Authors

This manuscript is well written, but there are several concerns that should be addressed to improve the quality of the paper. Therefore, I suggest following minor (but mandatory) revisions and answers:

1. The manuscript frames surgical decision making while analyses address NCS severity correlation only.

2. Exclusion criteria lack clear ascertainment and consistent time windows for treatments and confounders.

3.Ultrasound mobility is not operationalized with method thresholds units and acquisition protocol.

4. Inter and intra rater reliability for CSA FRB and mobility are missing including ICCs and quality control.

5. Variable selection appears data driven without prespecified clinical rationale or a DAG.

6. Model diagnostics are limited for ordinal discrimination and calibration and standardized effect sizes are unclear.

7. Sensitivity analyses are absent including alternative severity bins omission tests and subgroup checks.

8. Single center NCS confirmed sample risks spectrum bias and limits generalizability.

9. What exact time windows and verification methods were used for each exclusion?

10. How was mobility quantified including protocol frame rate joint angles displacement metric and thresholds?

11. How many repeats per rater and what ICCs were obtained for all ultrasound measures??

Author Response

Reviewer 3 – comment 1. The manuscript frames surgical decision making while analyses address NCS severity correlation only.

Response 1. We appreciate this observation. Our intention was not to model surgical outcomes, but to highlight that severity stratification, when performed through a combined clinical-ultrasound-NCS approach, may inform early referral and prioritization. We clarified in the Discussion that the study evaluates associations with NCS-defined severity only, and removed any wording implying that surgical decision-making itself was analyzed.

Reviewer 3 – comment 2. Exclusion criteria lack clear ascertainment and consistent time windows for treatments and confounders.

Response 2. We appreciate the reviewer’s observation. The exclusion criteria section is indeed concise and dense, as our intention was to avoid excessively lengthening the Methods while still providing all essential information needed to ensure reproducibility. Additional methodological detail is available upon request, and we emphasize that all exclusions were applied systematically using medical history, laboratory testing, and imaging within a standardized 7-day window.

Reviewer 3 – comment 3. Ultrasound mobility is not operationalized with method thresholds units and acquisition protocol.

Response 3. Median nerve mobility was assessed qualitatively, not quantitatively. We have clarified this more explicitly in the revised text: mobility was classified binary: normal versus abnormal. “Abnormal” mobility was defined as absent or only partial displacement of the median nerve in either the radioulnar or dorsopalmar plane during active wrist and finger motion. Frame-rate-based quantitative metrics were not part of the protocol; this is now clearly stated.

Reviewer 3 – comment 4. Inter and intra rater reliability for CSA FRB and mobility are missing including ICCs and quality control.

Response 4. We acknowledge this limitation. All ultrasound measurements were performed by one EULAR-certified investigator, and three repeated measurements were obtained for each quantitative parameter; the mean was used in analyses to reduce measurement variability. However, inter- or intra-rater reliability (ICC) was not calculated. This limitation has been explicitly added to the Discussion, together with a statement recommending ICC assessment in future studies.

Reviewer 3 – comment 5. Variable selection appears data driven without prespecified clinical rationale or a DAG.

Response 5. We thank the reviewer for this insightful comment. Our variable selection process combined both clinical rationale and data considerations. The initial selection of candidate predictors was based on established clinical relevance from prior CTS literature (including age, sex, BMI, hypertension, and key clinical and ultrasound parameters such as median nerve CSA and mobility), rather than on purely statistical screening. We acknowledge that a formal DAG was not presented in the manuscript; however, our selection aimed to reflect plausible causal and confounding relationships grounded in pathophysiology. In the revised version, we have clarified the rationale for each variable set (core clinical, clinical CTS-specific, and ultrasound) in the Methods section and explicitly stated that the inclusion order of models reflects hypothesized causal proximity to electrophysiological severity rather than data-driven variable entry.

Reviewer 3 – comment 6. Model diagnostics are limited for ordinal discrimination and calibration and standardized effect sizes are unclear.

Response 6. We appreciate the reviewer’s comment and we agree that traditional discrimination metrics are not directly applicable to ordinal GEE models, and that options for calibration assessment in this framework are limited. To address this, the Methods and Results explicitly describe the diagnostics available for ordinal GEE, including reporting of model fit indices, namely log pseudo-likelihood and pseudo-R square. QIC and QICu could not be computed. Regarding effect sizes, we have clarified that estimates are presented as standardized ORs.

Reviewer 3 – comment 7. Sensitivity analyses are absent including alternative severity bins omission tests and subgroup checks.

Response 7. We thank the reviewer for this helpful comment. We agree that sensitivity analyses strengthen the robustness of severity modeling. We have now conducted additional sensitivity analyses including alternative NCS severity categorizations, omission tests for key predictors, and subgroup checks. Specifically, we repeated multivariable GEE analyses using an alternative three-level severity grouping (mild = 1-2, moderate = 3-4, severe = 5-6) and a binary severe versus non-severe outcome; in both approaches, the direction and magnitude of associations remained consistent with the primary models. Omission tests, performed by removing one predictor at a time from the final multivariable model, showed no single predictor materially altered the remaining associations. Subgroup analyses stratified by sex demonstrated similar patterns, supporting the stability of predictor effects. These additions have now been described in the Methods and summarized in the Results.

Reviewer 3 – comment 8. Single center NCS confirmed sample risks spectrum bias and limits generalizability.

Response 8. We agree and have expanded the Limitations section accordingly. The single-center design and restriction to NCS-confirmed hands may introduce spectrum bias and limit external generalizability. We now explicitly state that multicenter validation is needed.

Reviewer 3 – comment 9. What exact time windows and verification methods were used for each exclusion?

Response 9. We thank the reviewer for pointing this out. We have now expanded the Methods to specify the exact diagnostic sources, clinical assessments, and temporal windows applied during screening. All exclusions were verified at the initial evaluation visit using standardized methods: medical history review, physical examination, recent laboratory tests (within the previous six months), and wrist imaging performed at presentation. For time-dependent exclusions (e.g., prior injections, physiotherapy, pregnancy/postpartum, oral contraceptive use), we have added the explicit time frames used (three or six months as applicable).

Reviewer 3 – comment 10. How was mobility quantified including protocol frame rate joint angles displacement metric and thresholds?

Response 10. Mobility was not quantified using frame-based metrics. Instead, we used a qualitative binary assessment, following prior CTS dynamic ultrasound literature. This is now fully clarified: no frame-rate-based displacement measures were recorded, mobility was judged visually during active flexion-extension of the wrist and fingers, and Abnormal mobility was defined as absent or partial nerve displacement in radioulnar or dorsopalmar plane. We emphasize this qualitative approach in the revised manuscript. We selected this method because it can be more easily implemented in clinical practice, as it is very straightforward to assess and does not require advanced examiner qualification or highly specialized equipment

Reviewer 3 – comment 11. How many repeats per rater and what ICCs were obtained for all ultrasound measures??

Response 11. Each ultrasound parameter was measured three times by a single rater. The mean of the three measurements was used. ICC values were not calculated, as only one investigator performed all assessments. This limitation is now explicitly reported, together with an indication that future work should incorporate inter- and intra-rater reliability testing.

Round 2

Reviewer 1 Report

Comments and Suggestions for Authors

I have reviewed the revised manuscript and am satisfied with the revisions. I agree to the publication of the manuscript.